# Temporal and Spatial Dynamics of Organ Water Content in Maize with Different Senescence Types

**DOI:** 10.3390/plants12183269

**Published:** 2023-09-14

**Authors:** Shang Gao, Bo Ming, Lu-Lu Li, Yi-Zhou Wang, Jun Xue, Peng Hou, Ke-Ru Wang, Shun-Li Zhou, Shao-Kun Li, Rui-Zhi Xie

**Affiliations:** 1Institute of Crop Sciences, Key Laboratory of Crop Physiology and Ecology, Chinese Academy of Agricultural Sciences, Ministry of Agriculture and Rural Affairs, Beijing 100081, China; g382824817@126.com (S.G.); obgnim@163.com (B.M.); lilulu19910818@163.com (L.-L.L.); xuejun5519@126.com (J.X.); houpeng811125@163.com (P.H.); wkeru01@163.com (K.-R.W.); 2College of Agronomy and Biotechnology, China Agricultural University, Beijing 100193, China; zhoushl@cau.edu.cn

**Keywords:** maize (*Zea mays* L), water content, spatiotemporal dynamics, stay-green trait, senescence

## Abstract

Understanding the water status of specific organs can be helpful in evaluating the life activities and growth conditions of maize. To accurately judge organ growth conditions and thus design appropriate interventions, it is necessary to clarify the true water dynamics of each maize organ. Using multiple maize cultivars with different growth periods, spatio-temporal water dynamics were analyzed here in the leaves, stalks, and ear components. Leaf water content was found to gradually decrease from both the bottom and top of the plant to the middle, whereas stalk water content decreased sequentially from the top to the bottom. Each successively higher node from the bottom of the plant was associated with decreases of 0.99% and 1.27% water content in the leaves and stalks, respectively. The water dynamics in leaves and internodes showed three clear stages: the slow loss, rapid loss, and balance stage. A water content of 60% appeared to be an irreversible turning point for initiation of senescence. Using normalized growth period as a measure, each of the tested cultivars could be assigned into one of two types based on their water dynamics: stay-water or general type. General-type cultivars had a shorter duration with a high water content and a water loss rate approximately twice as high as that of the stay-water type. This may have been related to the leaf senescence characteristics. However, the stay-water trait did not interfere with water dynamics of the ear components. Therefore, it may not be robust to evaluate the kernel dehydration of maize according to leaf senescence conditions due to the weak correlation between kernel water content and leaf senescence characteristics.

## 1. Introduction

Water is an essential substance for plant life and an important participant in plant metabolic activities such as photosynthesis [1]. The water content and water duration of a plant can be good indicators of growth condition. Maize (*Zea mays* L.), a critically important food and processing material, plays a key role in food security and industrial production [2]. Research into water-related activities in maize plants is of great significance in understanding plant growth status, which in turn enables appropriate management practices and intervention to maintain a strong output.

Maize water relations describe the water budget in the plants [3]. The water-driving force of transpiration and the water potential gradient formed by the soil–plant–atmosphere continuum together promote vertical water transport in plants [4]. However, water transport against the force of gravity is inherently difficult, and it is essential that plants remain sufficiently healthy to maintain the water drive. Stress or adverse conditions that endanger plant metabolism may alter the water status. Thus, water status dynamics are generally considered important indicators of plant health and environmental stress [3]. This principle has been widely applied in remote sensing to monitor plant health [5,6]. In previous studies, maize leaf water content has been used to monitor plant drought stress [7,8,9]. Declining leaf water content is closely related to decreases in stomatal closure and the maximum quantum yield of fluorescence [8]. An extremely low water status causes full hydraulic disruption and photochemical damage [10], indicating that the destruction of photosynthesis may be the last physiological symbol in living plants experiencing severe water stress. In maize, the top leaves was more vulnerable to stress due to their high position [9]. Differences in plant organs positions and the difficulty of vertical water transport suggest potential heterogeneity in the spatial distribution of water throughout a plant.

In addition to stress, the water statuses of plant organs are also affected by natural senescence. When leaf senescence is triggered by age information and other internal or external signals, the chloroplast disintegrates at first, followed by vacuole membranes disintegration, a loss of cytoplasmic membrane integrity, and cell death [11]. Finally, the leaves lose all physiological functions and begin to turn dry and yellow until they die completely. These changes in color and water status are easily perceived visually but it is the final stage of senescence. Thus, the decrease in water content can be regarded as the ultimate result of senescence. Due to the necessity of water for plant metabolism, rapid water withdrawal may indicate that the cells or organs have entered an irreversible stage leading to death. However, due to the difference in the activation of senescence-related genes in leaves, the senescence time will be changed, and the delayed leaf senescence will be reflected by the leaf color [12,13], often referred to as the leaf “stay-green” trait [14,15]. Correspondingly, stay-green leaves undergo a longer period of senescence, during which, they have a longer duration of high water content. Apparently, the senescence process is always accompanied by a change in water dynamics. There may be a synergistic relationship between water dynamics and senescence over time.

The plant stem, also known as the stalk, is an important supporting and conducting organ that also functions in photosynthesis [16,17]. Some studies have shown that the water content in basal maize stalk internodes slowly decreases during grain filling, weakening plant lodging resistance [18,19]. Although the basal internodes have the advantage of preferential water supply by the root system, they still lose water throughout the growth period. This suggests that the stalk water content is not constant during plant growth and development, even prior to senescence.

For monocarpic plants such as maize, the reproductive organ (ear) development often determines the timing of leaf and stalk senescence [20]. Mineral nutrients from stalks and leaves are often transferred to maize kernels in the process of grain filling [21,22], initiating programmed organ death [23,24]. Maize ears, which are independent reproductive organs, consist of kernels, husks, cobs, and peduncles. The kernels are the portion with direct economic value, and the kernel growth, development, water dynamics, and influencing factors have therefore been extensively studied [25,26,27].

Although some studies have addressed senescence and water dynamics in maize kernel, leaves, and stalks, little has been reported regarding the relationships between these components. It remains unknown whether there is a synchronous relationship between the water dynamics of maize reproductive organs and the water dynamics of leaves and stalks, and whether the water dynamics of maize ears are affected by those of stalks or leaves. Furthermore, it is unclear whether different positions of the leaves or stalks would exhibit similar or different water dynamics, and it is also unclear whether the leaves or stalks would have similar or different water dynamics compared to the ear. If the reproductive growth of maize is just a signal that triggers the senescence of leaves and stalks, there may be no time correlation in the water dynamics between these organs. Stalk and leaf senescence are inevitable physiological phenomena in monocarpic plants such as maize. Generally, differences in genetic background and environmental conditions lead to great variation in water status dynamics and senescence timing. However, because maize kernels are reproductive organs, their internal growth and development are typically regulated by a mechanism of homeostasis, particularly with respect to the water status [28], which is crucial to completing the plant life cycle. If normal kernel development could be affected by senescence, environmental disturbances (such as stress) or abnormal water conditions caused by genetic variations could have been fatal to the survival of the species. Therefore, we hypothesize that maize kernel water dynamics are independent and not influenced by other organs.

In the present study, we analyzed the water dynamics in the leaves, stalks, and ears of 12 maize cultivars with different growth periods. Notably, we separated each leaf and stalk internode after reproductive development to perform individual measurements of the water content at each position, which has rarely been conducted in previous studies. We then analyzed the water dynamics in the leaves, stalks, and ears over time and assessed the relationships between these factors. The objectives of our study were the following: (1) to clarify the vertical distribution of water content in maize leaves and stalks; (2) to decipher the water dynamics of each leaf and stalk internode during senescence; and (3) to evaluate the relationships between the water statuses of the stalks/leaves and the ears. This study was designed to shed light on the relationship between senescence and water dynamics in maize plants, particularly during organ senescence.

## 2. Results

### 2.1. Water Dynamics in Stalks and Leaves of Different Maize Cultivars

Contour diagrams of stalk water content (SWC) and leaf water content (LWC) were plotted based on interpolation results (Figure 1). In Figure 1a, the water loss of stalks all began at the top of the plant, then SWC gradually increased from the top to the bottom of the plant throughout the reproductive growth period. The SWC was above 75% in the basal internodes of almost all tested cultivars over the entire testing period. There were cultivar-specific differences in SWC, but in general, water loss was slower in cultivars with a long growth period. SWC remained above 60% around the ear at physiological maturity and was maintained at that level for a long period even after kernel maturation. This high water content showed that internodes at the ear position were maintained at a superior water condition until the filling was completed.

In contrast to the trends observed for SWC, the LWC was initially high in the middle of the plant but low at both the top and bottom (Figure 1b). LWC consistently decreased earlier at the bottom of the plant than at the top, and LWC started to decrease latest in the middle of the plant. The latest leaves to senesce were consistently near the ear (either the ear leaf or one to two leaves above the ear leaf). This phenomenon meant that leaf senescence began at both ends of the plant and progressed through a strict sequence. In addition, LWC eventually stabilized at every position, but the time to stabilization varied between cultivars. Generally, the cultivars with a long growth period required a longer time (larger green area in Figure 1) to stabilize than those with a short growth period.

### 2.2. Vertical Distributions and Relationships of Stalk and Leaf Water Content

The water dynamics of each internode and leaf position were analyzed in four maize cultivars with different growth periods: the early-maturing Hetian1, the middle-maturing Dika517 and Zeyu8911, and the late-maturing Zhengdan958. The data of four special stages (the maize kernel filling stage (Filling), kernel physiological maturity (Maturity), ~15 days after maturity (AM15), and ~30 days after maturity (AM30)) were shown in Figure 2. At Filling, SWC and LWC were nearly equal and were maintained around at 80–90%; only the several basal leaves showed a significant decrease in LWC. At Maturity, LWC was consistently significantly lower than SWC at the same position, except in the middle leaves; the basal LWC dropped to about 10%. These extremely low water content levels indicated that these leaves have completely died. All of the leaves in cultivars with a short growth period (Hetian1 and Dika517) were completely dead at AM15, whereas the leaves of the other cultivars did not die until AM30. However, the observed differences in water duration could not be fully explained by cultivar-specific differences in the growth period, because the water duration of the middle-maturing Zeyu8911 was much longer than that of the middle-maturing Dika517, and longer than that of the late-maturing Zhengdan958. Delayed leaf senescence caused by cultivar-specific characteristics may have been an important reason for the differences in water duration. Decreases in the mean SWC and LWC were also associated with increases in the standard deviations of these measurements. This may have been related to individual variations in the time to death among each population.

In terms of the patterns associated with water loss, decreases in LWC began at the top and bottom of the plant, but all positions lost water simultaneously after Maturity. In contrast, SWC decreased sequentially from the top to the bottom of the plant. Only when the uppermost internode died, did the next internode started to die. This may have been due to the necessity of maintaining water flow through the lower stalk internodes. In addition, there was a priority order for the decrease in LWC and SWC at the same position. In the upper part, LWC always consistently decreased earlier than SWC did at the same position. Considering the necessity for the transport of nutrients and water from the stalks to the leaves, the rules of water loss were as follows: leaf death was a prerequisite for stalk death, but upper stalk death was necessary for subsequent internode death.

Water conditions were not uniform between positions. Both the SWC and LWC declined gradually (Figure 3), even in the vigorous stage of reproductive growth (Filling). We further analyzed the spatial and vertical distribution characteristics of SWC and LWC in all of the tested cultivars. During Filling, both the SWC and LWC showed a steady decreasing trend from the bottom to the top of the plant within a range of 90–60%. There were no clear differences between cultivars. At Filling, LWC decreased ~1.27% between each position from the bottom to the top of the plant. This was slightly larger than the decrease in SWC between each internode in the same direction (0.99%). At Maturity, SWC decreased by an average of 0.70% at each internode; this increased to 1.14% at AM15. The vertical distribution of LWC was inconsistent at these later stages due to the rapid water loss. Additionally, the SWC of each living internode was not constant and slowly decreased during the reproductive growth period. The average SWC of the 10 basal internodes decreased by more than 5% from Filling to Maturity (data are not shown). 

Notably, the SWC and LWC stably declined in all tested cultivars when they were above 60%. However, once the water content dropped below 60%, the water loss rate increased sharply, showing a cliff-like decline. This meant that these stalks or leaves may be undergoing some drastic changes, such as rapid death. Therefore, we speculated that the threshold of 60% water content may have particular physiological significance for stalks and leaves, and that this is a critical threshold for the maintenance of physiological functions during senescence.

### 2.3. Temporal Dynamics of Stalk and Leaf Water Content

The temporal water dynamics were analyzed for each stalk internode and leaf, and the water dynamics of four maize cultivars are shown in Figure 4 (results of other tested cultivars are shown in Appendix A). At all positions and in all maize cultivars, the SWC and LWC first decreased slowly, then decreased rapidly, and finally reached a stable state. Basal internodes maintained a high SWC throughout the test period, and the whole water dynamics was only shown in the upper internodes. Almost all LWC levels dropped to a stable value (about 10%), although the required time varied by position. In addition, both the stalks and leaves entered the rapid stage of water loss once the water content was reduced to about 60%, which was consistent among cultivars. As discussed above, a 60% water content appeared to be an important threshold dividing slow and rapid water loss, and may have been the lowest point at which physiological functions could be maintained. Rapid water loss from an organ suggests that the constituent cells can no longer hold water steadily. Subsequently, when an organ has died, the water content gradually reaches a dynamic balance with ambient water levels, resulting in a stable and very low apparent water content.

Based on these results, we roughly divided the temporal dynamics of SWC and LWC into three stages: Stage I (slow loss stage); Stage II (rapid loss stage); Stage III (balance stage). There were two important turning points between each two stages. Figure 5 shows a schematic diagram of the three stages, and each stage was fitted with a linear model. Line AB represented Stage I, line BC was Stage II, and line CD was stage III. The equations were as follows: $$\left\{\begin{array}{c} \mathrm{Stage}\;I:\;\;\;\;y=k_{1}\cdot x+b_{1}\;,\;\;\;\;\;\;\;\;A<x\ll B\\ \mathrm{Stage}\;\mathrm{II}:\;\;\;y=k_{2}\cdot x+b_{2}\;,\;\;\;\;\;\;\;B\ll x\ll C\\ \;\;\mathrm{Stage}\;\mathrm{III}:\;\;\;y=b_{0}\;\;\;\;\;\;\;\;\;\;\;\;\;\;\;\;,\;\;\;\;\;\;\;\;\;\;\;\;\;\;\;\;x\gg C \end{array}\right.$$
where *k*_1_ and *k*_2_, which represent the rate of water loss, were the slopes of the fitted equations (0 > *k*_1_ > *k*_2_). *b*_1_ and *b*_2_ are the intercepts of the equations. *b*_1_ is related to initial water content, and the *b*_2_ has no special physiological significance. Point *B* is the turning point from slow loss stage to rapid loss stage, which represented the water content threshold of irreversible organ death. Point *C* is the balance (*b*_0_) threshold, after which the organ water content is consistent with the water content of the surrounding environment. The X-axis represents a time course, such as days after sowing, days after silking, or other time series. The Y-axis represents the water content of the stalks or leaves. 

According to our study, during Stage I, the initial SWC and LWC were 80–90% at the bottom of the plant and 70–75% at the top. The slope of the decline was about 0.99% in the stalks and 1.27% in the leaves at Filling (Figure 3). The water threshold from Stage I to Stage II was about 60%. The water content at Stage III was about 10%, but depended on the environmental conditions of each test site. Due to the different characteristics of the maize cultivars, the days required for water loss varied in each stage.

### 2.4. Classification of Water Dynamics in Stalks and Leaves

To further simplify and analyze the organ water dynamics in different cultivars, it is necessary to control for cultivar-specific differences in the maize growth period. Therefore, the X-axis of water dynamics was converted to the Normalized Growth Period (NGP), and the changes in the average SWC and LWC of the whole plant before and after the use of NGP are shown in Figure 6. 

When using days after silking, most of stalks did not show significant water loss during the whole growth period, and the average SWC showed only a slight decrease. Due to the rapid death of almost all leaves, the LWC gradually declined at a slow rate to rapid rate. In addition, in Figure 6a,c, there were large water variations due to differences in the growth period, and it was difficult to summarize a regular water dynamics process. However, the water dynamics became clearer and more convergent when the growth period was expressed as NGP, especially in leaves (Figure 6b,d). The leaf water dynamics were clearly divided into two types. The first type had a fast LWC loss rate and was named as the general type in this paper, and included nine maize cultivars: Hetian1, Fengken139, Yudan132, Dika517, Jingnongke728, Zhengdan958, Xianyu335, Dika653, and Xianyu1331. The second type had a slow water loss rate, and was named the stay-water type, and included three maize cultivars: Yuanhua9, Zeyu8911, and Liaodan585. The water loss rate of the general type was about twice than that of the stay-water type. The water loss rate only began to differ around physiological maturity (NGP = 1), and there was almost no significant difference before maturity. The difference did not appear to be related to the growth period of the cultivars. For example, the traditional late-maturing cultivars Zhengdan958 and Xianyu335 showed larger water loss rates than the middle-maturing Zeyu8911 (growth periods are shown in Table 1).

The analysis of leaf water dynamics at each position showed that the leaf stay-water characteristic primarily manifested above the ear, around the top 8 to 10 leaves of each plant (Figure 7). Most of the lower leaves did not show significant differences in water loss, which may have been due to the comparatively earlier death of these leaves. In addition, the stay-water characteristics of stalk internodes at different positions were also significantly different among cultivars. The patterns were nearly identical to the patterns in leaves (Appendix A). However, the differences in stay-water characteristic were smaller in the stalk internodes than that in leaves due to the delayed water loss observed in many internodes.

Although the different types of maize cultivars varied in the water duration and loss rate, they all followed the same three-stage water loss process (although Stage III was not observed in the stay-water cultivars). The two types only differed in the water loss rate and the timing of the turning point. Therefore, the water dynamics of the upper leaves for both types were fitted using the water dynamics equation, and detailed information of the fitting parameters can be found in Appendix A. Since most of stalk internodes did not show the complete water dynamics process during the test period, no special discussion was included in this paper. The detailed information of the stalks can be found in Appendix A.

The special parameters of leaf water dynamics in the two types are shown in Figure 8. When the LWC dropped to Point *B* (turning point from Stage I to Stage II), there was roughly a parabola-shaped trend with the decrease in leaf position, and stay-water cultivars appeared later. However, differences between the two types were primarily observed in the second to fifth leaves, and the values for the other leaves were comparable between types. In addition, the LWC at turning point B was similar for both stay-water and general types, and was in the range of 60–70%. Based on the results of the fitted equations, there were significant differences between the parameters of the two types. The average value of *k*_1_ and *k*_2_ were −23.00 and −142.68, respectively, for the general type, but were −13.01 and −83.83, respectively, for stay-water type. The initial LWC of living leaves (*b*_1_) was same and the average was about 84.76% for both types.

### 2.5. Water Dynamics of Ear Organs and Their Relationship with Ear Leaves and Ear Stalk Internodes

Figure 9 and Figure 10 show the water dynamics of ear organs and its relationship with ear leaves/internodes. Figure 9 shows that the kernel water content decreased monotonically throughout the growth period. The water dynamics of cob and husk were similar to those of leaves or stalks, with the water loss process comprising three stages. At physiological maturity (NGP = 1), the husk water content was already in Stage III, whereas cobs had just entered Stage II. Of the four ear components, the water content was highest and water loss was lowest in the peduncle.

The kernels and ear internodes/leaves had different water loss patterns, and the water content was the lowest in the kernels (Figure 10a). The husks and ear leaves had a similar initial water content which may be related to similarities in structural properties. However, the husk water dynamics had a shorter Stage I and entered into Stage II earlier compared to the ear leaves (Figure 10b). The cob had a significantly lower water content and entered Stage II earlier than the ear internode during the test period (Figure 10c). The water content in the peduncle was similar but had a slightly higher value (about 4.08%) compared to that of the ear internodes (Figure 10d), meaning that the peduncle, a part of the reproductive organ, had superior water conditions compared to the stalk.

Although the different maize cultivars had two stay-water characteristics, the fitted curves of the water dynamics in kernels, husks, and cobs completely overlapped between the two types, and there were no significant differences between cultivars. Differences in stalks and leaves water dynamics due to stay-water characteristic did not alter the water dynamics of these organs. Therefore, the maize ear (except for the peduncle), the most important organ for reproduction and yield, may have a unique water dynamic that is independent of the stalk and leaf water dynamics.

## 3. Discussion

As a typical tall-stalk crop, maize plants consist of many stalk internodes and single leaves (including sheaths) attached to the nodes and an additional reproductive organ (ear) that grows on a node in the middle. Due to differences in developmental initiation and lifespan or under environmental stress, stalk internodes/leaves at different positions have different water conditions [29], especially during reproductive growth stages. Because maize is a typical monocarpic plant, reproductive development is accompanied by programmed senescence of the other organs [23,24]. During the process of organ senescence, water loss is the most easily detected signal; for example, leaf death is accompanied by obvious drying. In this study, the leaf water content (LWC) and stalk water content (SWC) decreased continuously during reproductive growth stages. The LWC decreases began at both ends of the plant and progressed to the middle, whereas SWC declined sequentially from the top of the plant due to the necessity for water and nutrient transport at the lower part of the plant. The middle leaves (one to two positions above the ear) and the bottom internodes had a longer duration of high water content, meaning they experienced a longer senescence time. These results were largely consistent with prior studies showing that the middle maize leaves have the longest functional duration [30,31].

In the vertical direction, the SWC and LWC had similar patterns: the water content of surviving leaves/internodes decreased gradually from the bottom to the top of the plant. The vertical water gradient may be related to water potential caused by the soil–plant–atmosphere continuum [4], which has also been found in sugarcane stalks [32]. In the early stage of ear development (Filling stage), the water content of living leaves and internodes decreased by 1.27% and 0.99%, respectively, at each position from the bottom to the top of the plant. As the ear developed, the SWC of every lower internode showed a decreasing trend, suggesting that even living stalks still lost water during reproductive growth. The water loss of living stalks may be due to the decreases in stalk physiological function. Or, it may be caused by the difficulty for maize plants to maintain adequate transpiration and a high water-driving force as a result of the massive leaf death [4]. Additionally, at each position, the LWC consistently declined prior to decreases in SWC. Based on the principle of “nearby transportation”, the water and nutrients transported by the stalks will be preferentially supplied to the adjacent leaves. However, once the leaves die, the exchange relationship would disappear. In order to reduce extra energy consumption caused by functional redundancy, triggering programmed death may be the best solution. Therefore, the top-to-bottom sequence of water loss in stalks may be related to functional redundancy resulting from leaf senescence. This sequential senescence was not observed in the bottom stalk internodes because it was necessary to maintain water and nutrient transport to the living organs.

Although the senescence time in maize leaves can be affected by some management methods or interventions such as fertilizer and plant density [33,34], the order and results of leaf senescence cannot be changed [35]. The unalterable rhythm of senescence suggested that senescence-associated water loss may also have some similar rhythms that can be explored. The changes in physiological functions during senescence were similar, such as chlorophyll disintegration, membrane system destruction, and drying [11,20]. When the decline in the LWC can be easily perceived, nutrient transfer and functional recession have been completed and senescence is in its final stage. Correspondingly, water dynamics during senescence may follow specific stages. Indeed, we found that the water dynamic of maize leaves could be divided into three main stages. The three stages could be fitted by linear equations, and the slopes of the equations represent the water loss rate. In Stage I, the leaves’ function continued to decline but still maintained sufficient vitality. At this time, the leaves were in the early stage of senescence, and the water loss was slow and slight. However, the low rate of water loss meant that the leaves were getting older and matter transfer had already begun. Thus, the equation intercept at this stage may be related to the LWC at the beginning of senescence (initial value of leaf water loss). As senescence continued, programmed disintegration of organelles exacerbated functional recession. In particular, after the disintegration of the membrane system, the cells gradually lost their water storage capacities and began to lose a lot of water. At that time, the water loss entered a rapid stage (Stage II), which symbolized irreversible death [11,36]. There was an important turning point in the transition from Stage I to Stage II. The LWC at this turning point appeared to be a stable value, with the lowest threshold around 60% (Figure 4 and Appendix A). The abnormal leaf function and photosynthesis caused by water stress follows a specific timeline. At the initial stage of leaf water loss, the stomata opening decreases and gas exchange resistance increases, but stomatal pores reopen and photosynthesis returns to the original level after rehydration [8,37]. However, with the intensification of stress, plant oxidation and reactive oxygen species continue to be generated, resulting in the destruction of chloroplast structure and membrane systems [38]. Meanwhile, photosynthetic pigments degrade and destroy the photosynthetic electron transfer system, ultimately leading to the damage of the photosynthesis machinery [39]. The decline in photochemistry of photosystems was affected only at a much lower leaf water content [10]. A previous report identified 68.26% LWC as the lower limit for photosynthesis in maize leaves after long-term drought stress [9]. Our results were similar to this study, suggesting that the threshold of LWC may be the minimum water requirement for maize leaves to complete life activities. The lower value in our study may imply complete irreversibility of the death process. During Stage III, the leaves were comparable to maize straw, and the LWC depended on the temperature and humidity conditions of the external environment and its own structural characteristics, and enter a balance with the water in environment. Other studies have reported this balance between environmental water levels and those of maize straw [40]. Furthermore, maize stalks actually follow a water dynamics process similar to that of the leaves. However, during the entire growth period (about 100 days after silking in this study), only a few of the upper internodes had significant water changes, and the SWC of many basal internodes only had a slight decrease. If longer times are tested, the complete water dynamics of the stalks may be observed.

In this study, the normalized growth period (NGP) was used to evaluate the water dynamics of stalks and leaves. In contrast to analyses measuring the growth period in terms of days after silking, the NGP could clearly divide the water dynamics of different cultivars into two types, which greatly reduced the difficulty of predicting water dynamics of different maize cultivars. Therefore, the normalization method may have potential application value for evaluating the stay-green or stay-water characteristics in maize cultivars. For maize leaves, the leaves of the stay-water type had a longer duration of high water content (a longer period in Stage I), indicating that the leaves of this type senesced later than those of the general type. However, this phenomenon was not due to the growth period of the maize cultivars, but may be related to their stay-green characteristic. The stay-green cultivars typically had longer leaf functional durations, leading to a delay in senescence and water loss. From this point of view, the stay-water leaf characteristic was a direct result of the stay-green characteristic, and the stay-green and stay-water characteristics are essentially equivalent. In addition, the water loss rate of the general type was almost twice that of the stay-water type in Stage II. It can be seen that the maize cultivars of the stay-water type (also stay-green type) not only delayed the senescence time of leaves, but also had slower senescence rates than that of general types. Most studies on maize stay-green characteristics focused on the leaves [12,14,21,41]. However, our results suggested that the stay-green or stay-water characteristics of stalks and leaves are actually synchronized, that is, when the leaves of maize cultivars have stay-green or stay-water characteristics, the corresponding stalks also have stay-green or stay-water characteristics.

The selection of stay-green cultivars plays an important role in the improvement of maize yield, because the delayed leaf senescence greatly enhances the post-flowering photosynthetic production capacity [15,42,43,44]. However, we did not find that the stay-green (stay-water) characteristic affected the dehydration process of maize kernels (Figure 10). The water content of ear organs (kernels, cob, and husk) of maize had inherent water dynamics, which was independent of the LWC and SWC. Previous studies have shown that the maize kernel had a high internal homeostasis and rhythm during reproductive growth stages [25,45]. This homeostasis allowed the kernel and its water to remain stable even under stress, during competition, and in limited physical growth spaces [28]. These findings may be helpful to explain why the water dynamic of kernels were not affected by those of leaves. However, the associated mechanisms require further exploration. It does not appear that kernel water content could be robustly predicted based on physiological indicators that are closely related to maize leaf senescence characteristics (such as chlorophyll content [46]). Therefore, in the selection of maize cultivars, the selection based on dehydration performance should pay more attention to the maize kernel itself, and evaluating dehydration performance or predicting water dynamics of maize kernels according to the stay-green or stay-water characteristics may lead to some misjudgments.

## 4. Materials and Methods

### 4.1. Experiment Design

Field data were collected from two different maize planting regions (spring and summer maize) in China in 2018. The spring maize planting area was the Western Agricultural Research Center of Chinese Academy of Agricultural Sciences (CAAS) in Changji, Xinjiang (N35°18′, E113°54′), and the summer maize planting area was the Comprehensive Experimental Station of the CAAS in Xinxiang, Henan (N35°18′, E113°54′). A total of 12 maize cultivars with different growth periods were tested. Nine maize cultivars were planted in Xinxiang. Each cultivar was planted in a plot of 25 m × 7.2 m at a density of 75,000 plants/hm^2^. Three maize cultivars were planted in Changji in plots of 100 m × 6 m, at a density of 90,000 plants/hm^2^. In both regions, plants were spaced 0.6 m apart. Plots for each cultivar were arranged at random. Field management was consistent with local practices. Emergency dates and silking dates were recorded for all cultivars. The date of physiological maturity was classified as the day on which the milk line disappeared and a black layer of kernels appeared (Table 1).

### 4.2. Organ-Specific Water Content Measurements

Healthy plants with similar growth conditions were selected for sampling. Samples were taken at intervals of 5–7 days from 3–5 plants in a time. The sampling start time was 21–31 days after silking in Xinxiang and 40–50 days after silking in Changji. The entire aboveground portion of each plant was sampled. The plants were cut at each stalk node to divide them into single-stalk internodes and leaves in order. Stalk internodes and their adjacent leaves (together with leaf sheaths) were deemed as the same positions. These portions were numbered in order from the top to the bottom of each plant. The tassel and corresponding internode at the top of plant was classified as the first stalk internode, stalk (0), and correspondingly, its adjacent leaf at the top of the plant was designated leaf (0). Lower stalk internodes and leaves were sequentially named, starting with stalk (−1) and leaf (−1). The ear was completely removed and the positions was recorded. The husk, cob, peduncle, and kernels were then separated for water content measurements, as described below. The judgment of the ear leaf/internode of maize cultivar was based on the frequency of ear appearance in samples in this study (Appendix A). The ear positions of the tested maize cultivars ranged from (−5) to (−8).

After maize organs were fully separated, the fresh weight of each position was measured immediately to reduce water loss. The dry weight were measured after drying at 105 °C for 30 min and then drying at 85 °C for at least 48 h in an air oven. The calculation of water content was as follows: Water content (WC, %) = (Fresh Weight − Dry Weight)/Fresh Weight × 100%. One hundred kernels in the middle of the ears were used to determine the kernel water content. For all other tissues, water content was measured using the entire sample. Subsequent analyses were conducted using the average water content from three to five replicates from each position.

### 4.3. Calculation of Normalized Growth Period

To control for cultivar-specific differences in the growth period, the Normalized Growth Period (NGP) was calculated in this study. The NGP was defined as the ratio of the number of days from sampling to silking over the number of days from silking to physiological maturity:NGP = (Sampling Date − Silking Date)/(Physiological Maturity Date − Silking Date)
where NGP is the Normalized Growth Period (decimal). At the silking date, the NGP is 0. When maize kernels reach physiological maturity, NGP is 1. After physiological maturity, NGP is >1.

### 4.4. Data Processing and Analysis

The R v4.1.3 was used for data smoothing over the sampling period. Water content was interpolated using two variables (days after silking and organ position) to smooth data with an interval of 1 day using Locally Weighted Regression (LOESS). The smoothing results were only used for the contour diagrams of water content. Microsoft Excel 2013 was used for data processing, and SAS 9.40 software was used to map and perform other statistical analyses such as line or curve fitting.

## 5. Conclusions

This study served to clarify the temporal and spatial dynamics of water content in maize stalks and leaves. Spatially, the water content of maize leaves and stalks showed a decreasing vertical water gradient from bottom to top (stalk and leaf decreased 0.99% and 1.27% per position at the filling stage, respectively). In the time course, the water dynamics of the leaves and internodes followed a specific order: in the leaves, water loss began at the bottom of the plant, then spread to the top, and finally to the middle. In contrast, in the stalk internodes, the water content sequentially decreased from the top to the bottom of the plant. The water dynamics of leaves and stalks could be divided into three main stages: the slow loss, rapid loss, and balance stage. The shift from the slow loss to the rapid loss stage represented an important threshold: the irreversible turning point leading to organ death. This was associated with a minimum threshold of about 60% water content. In addition, the normalized growth period allowed separation of maize organ water dynamics into two types, the stay-water type and the general type, which greatly reduced the difficulty in predicting the water dynamics in different cultivars. Stay-water cultivars had a longer water duration and delayed senescence time compared to general cultivars. During the rapid loss stage, the water loss rate of the general type was approximately twice as high as that of the stay-water type. The different types did not precisely correspond to cultivar-specific differences in the maize growth period, but may have been related to the delay in leaf senescence among stay-green maize cultivars. However, the stay-water characteristics of stalks and leaves did not affect the water dynamics of kernels, husks, and cobs in maize ears. Therefore, in breeding maize cultivars, it may not be reliable for evaluating the kernel dehydration by the stay-green characteristics of the leaves. These results are expected to be useful for monitoring maize growth in the late growth stage and for building and improving maize growth models and parameters, particularly for research on maize senescence.

## Figures and Tables

**Figure 1 plants-12-03269-f001:**
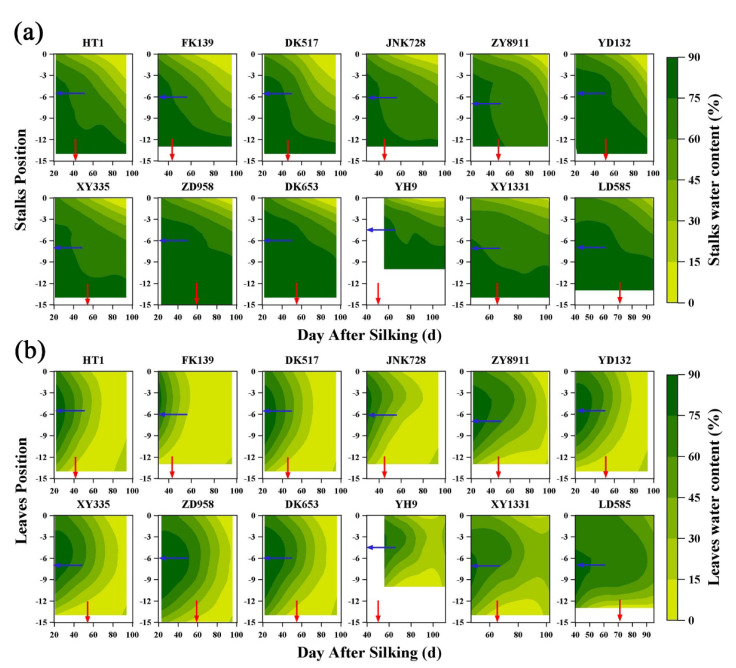
Contour diagrams of the relationship between stalk (**a**) and leaf (**b**) water content by days after silking of different maize cultivars. The blue and red arrows represent the location of the maize ear and the physiological maturity, respectively. Maize cultivars are arranged by the length of growth period from left to right. The abbreviation (full name) of maize cultivars names are FK139 (Fengken139); HT1 (Hetian1); JNK728 (Jingnongke728); DK517 (Dika517); XY335 (Xianyu335); ZD958 (Zhengdan958); ZY8911 (Zeyu8911); YD132 (Yudan132); DK653 (Dika653); YH9 (Yuanhua9); XY1331 (Xianyu1331); and LD585 (Liaodan585).

**Figure 2 plants-12-03269-f002:**
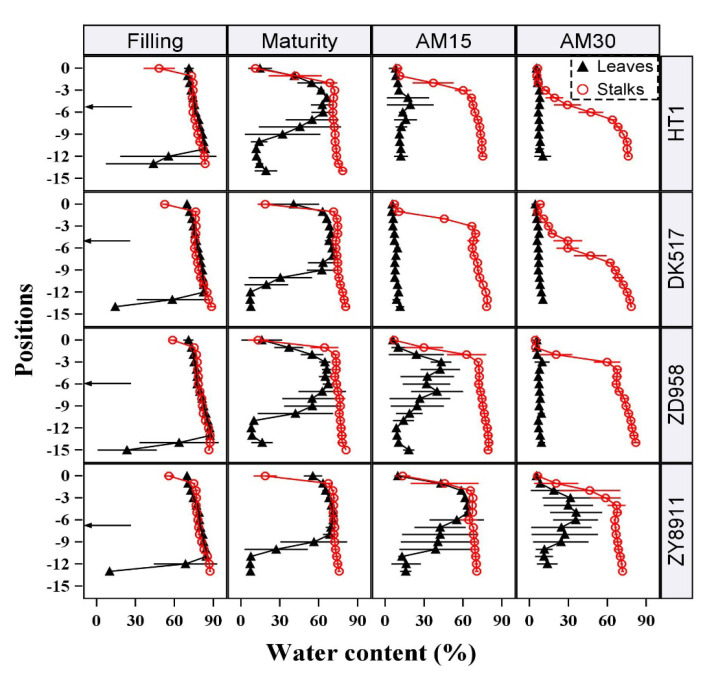
Water conditions of stalks and leaves at different growth stages of four representative maize cultivars. The black arrows on the left represented the approximate location of maize ear. “Filling” represents the filling stage of maize kernels (about 20–30 days after silking), “Maturity” represents the physiological maturity of maize kernels, and “AM15” and “AM30” represent 15 and 30 days after physiological maturity, respectively. The symbols with a line represent “mean value ± sd”. The maize cultivars names are Hetian1 (HT1), Dika517 (DK517), Zeyu8911 (ZY8911), and Zhengdan958 (ZD958).

**Figure 3 plants-12-03269-f003:**
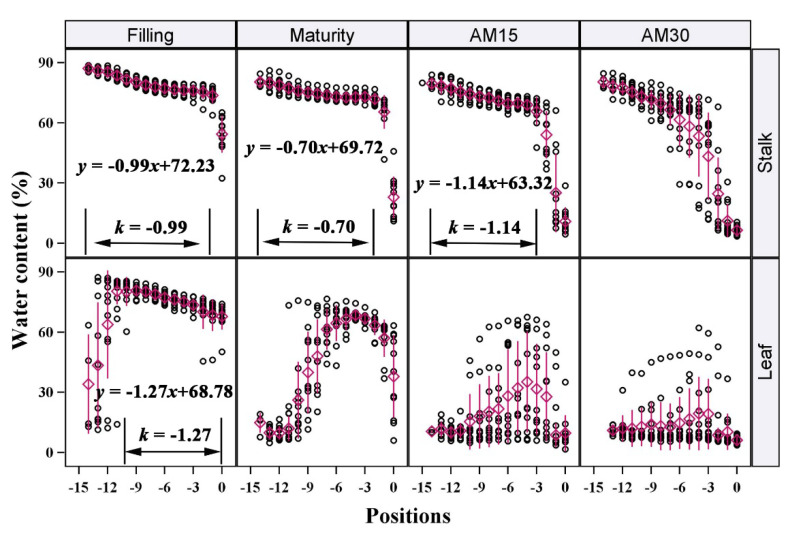
Vertical distribution of water content of stalks and leaves of maize at different growth stages. Each position contained all data for 12 maize cultivars. The k is the slope of the linear fitting equation of data between two black lines. The purple diamonds with a line are the “mean value ± sd”. The meanings of “Filling”, “Maturity”, “AM15”, and “AM30” are the same as in Figure 2.

**Figure 4 plants-12-03269-f004:**
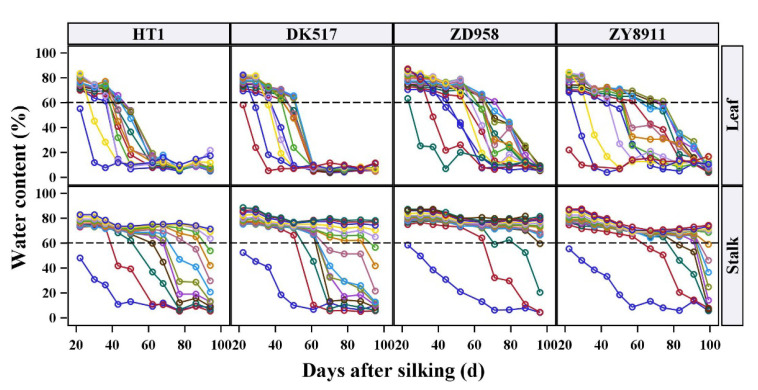
Water temporal dynamics of each stalk internode and leaf of four maize cultivars. The lines with different colored circles represent the changes of stalk internodes or leaves water content at different positions, and the same color in different figures represents the internodes or leaves with same position. The dash line is the reference line of 60% water content. The maize cultivars names are the same as in Figure 2.

**Figure 5 plants-12-03269-f005:**
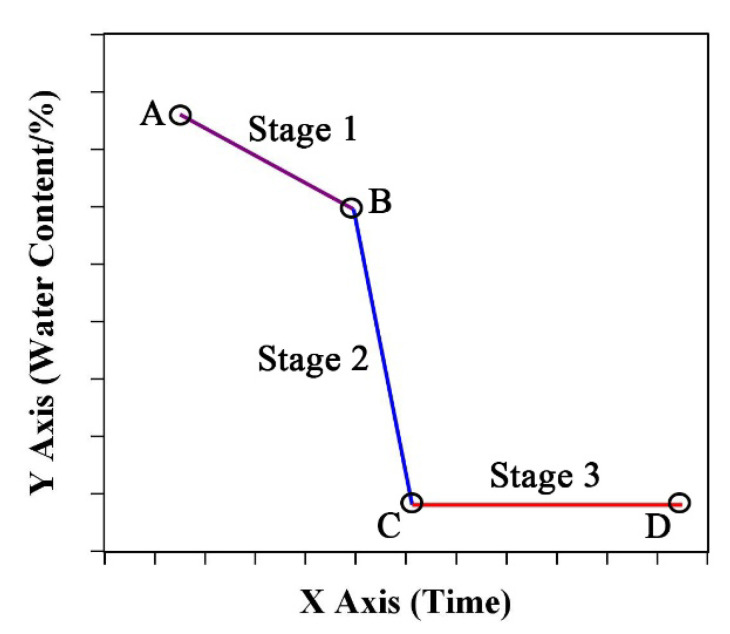
Schematic diagram of water temporal dynamics of stalks or leaves during growth period of maize. The A and D represented the initial and end points of the test respectively. The B and C represented the two turning points of the water content.

**Figure 6 plants-12-03269-f006:**
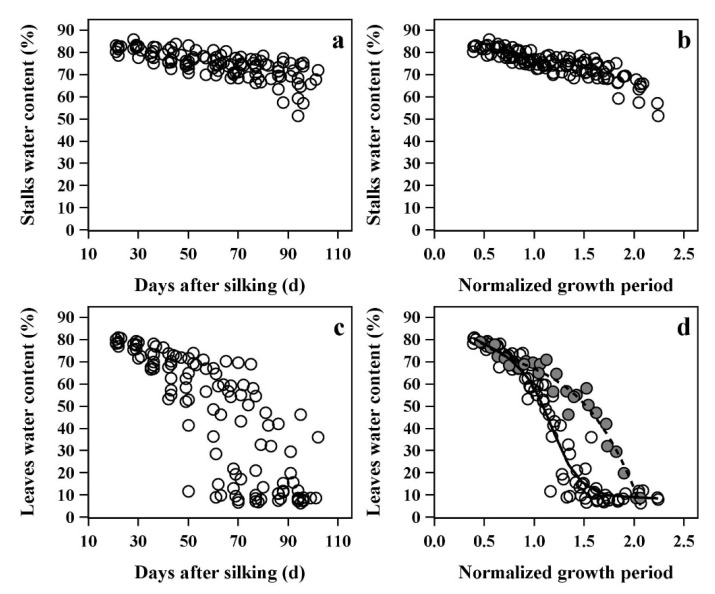
Relationship between average water content of all stalks and leaves in plants by days after silking (**a**,**c**) and normalized growth period (**b**,**d**).

**Figure 7 plants-12-03269-f007:**
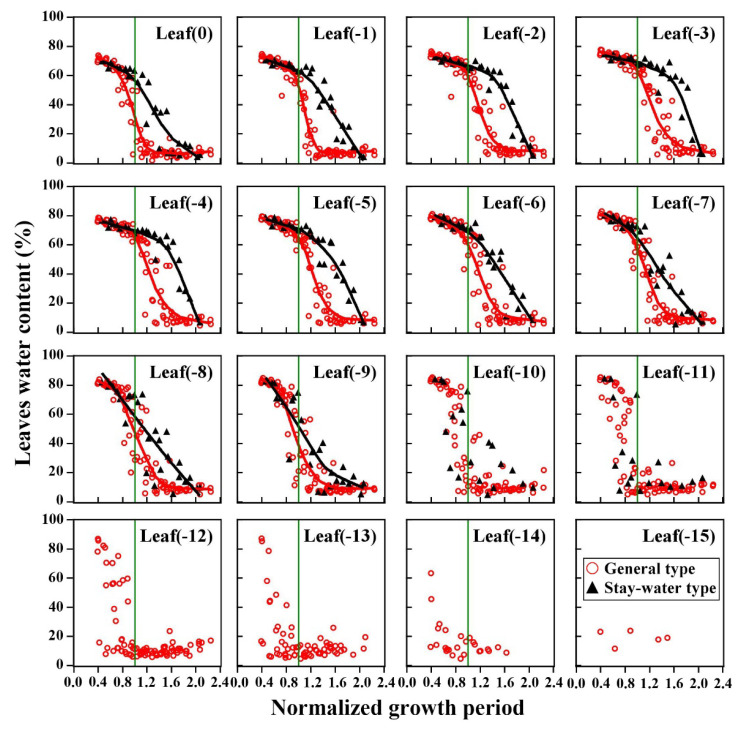
Differences in leaf water content of different types at different positions. The colored lines represent the LOESS (Locally Weighted Regression) fitted lines for the data.

**Figure 8 plants-12-03269-f008:**
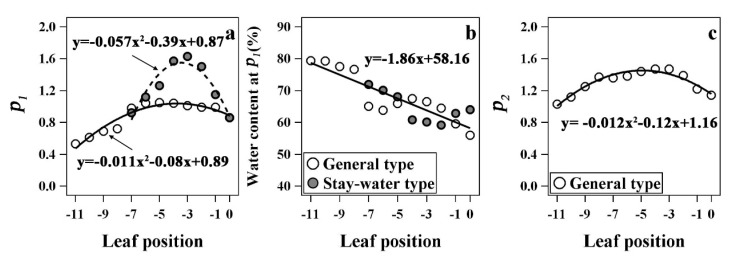
Comparison of the two turning points (**a**,**c**) and the water content at first turning point (**b**) of leaf water dynamics equations of different types. The curves in (**a**,**c**) were fitted by a quadratic polynomial, and (**b**) is a linear regression. *P*_1_ and *P*_2_ are the first and the second turning points of the water dynamic equation.

**Figure 9 plants-12-03269-f009:**
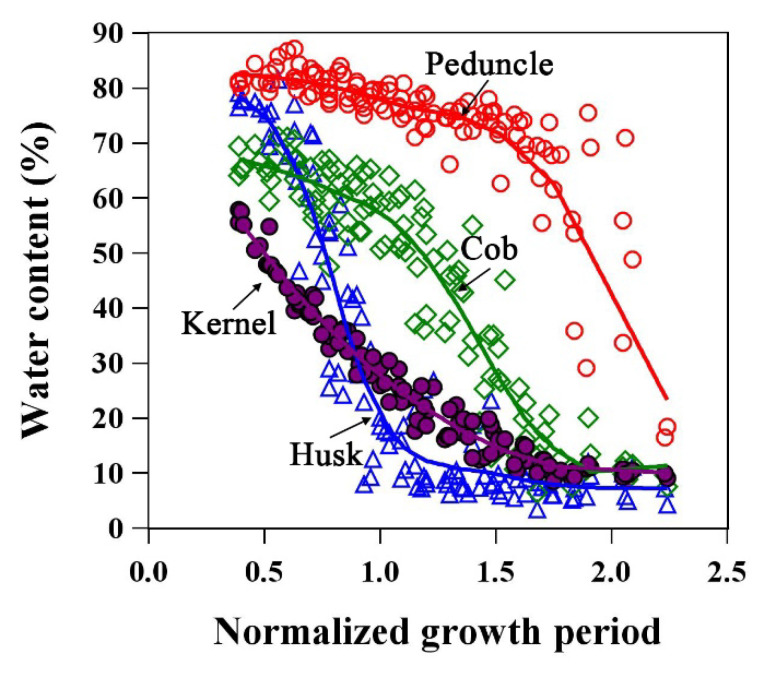
Water dynamics of kernel, husk, peduncle, and cob in maize ear. The lines are LOESS fitted lines.

**Figure 10 plants-12-03269-f010:**
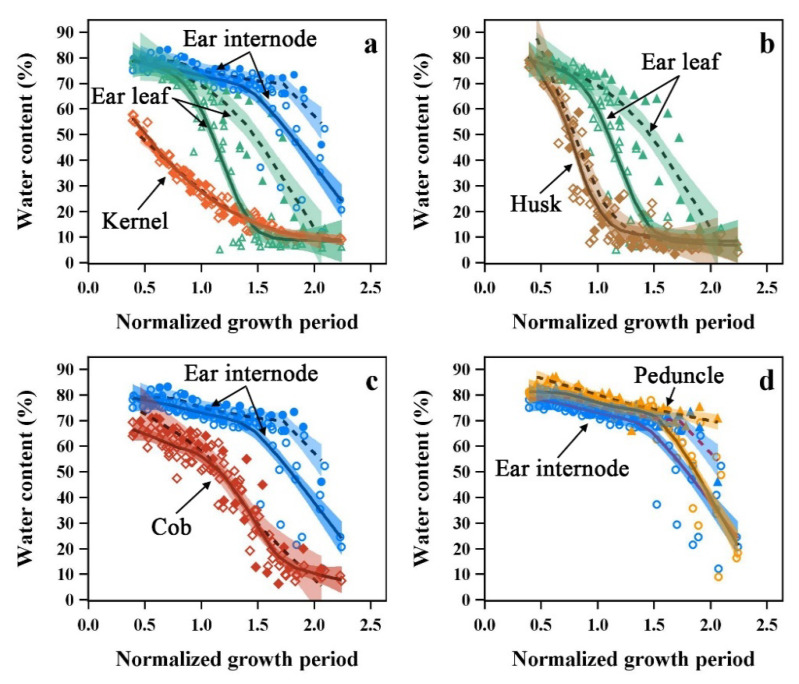
Relationships between kernel (**a**), husk (**b**), cob (**c**) and peduncle (**d**) of maize ear organs and ear leaves and ear internodes of the two types. Different symbols with colors represent different organs, and ear leaf and ear internodes use the same symbols and colors in the figure. Stay-water type (filled symbols) and general type (no fill). Solid lines and dash lines are the LOESS fitted lines of general type and stay-water type, respectively, and colored bands are the 95% CLM of the fitted lines.

**Table 1 plants-12-03269-t001:** Detail information of tested maize cultivars.

Site	Cultivar Name	Sowing Date (D. M)	Emergency Date (D. M)	Silking Date (D. M)	Physiological Maturity Date (D. M)	Growth Period (d)
Xinxiang, Henan	Fengken139	3 June	8 June	20 July	1 September	90
	Hetian1	3 June	8 June	21 July	1 September	90
	Jingnongke728	3 June	8 June	23 July	6 September	95
	Dika517	3 June	8 June	23 July	7 September	96
	Xianyu335	3 June	8 June	25 July	17 September	106
	Zhengdan958	3 June	8 June	26 July	21 September	110
	Zeyu8911	3 June	8 June	23 July	8 September	97
	Yudan132	3 June	8 June	25 July	14 September	103
	Dika653	3 June	8 June	26 July	19 September	108
Changji, Xinjiang	Yuanhua9	1 May	6 May	30 June	19 August	110
	Xianyu1331	29 April	6 May	9 July	12 September	136
	Liaodan585	28 April	6 May	16 July	25 September	150

## Data Availability

The data that support the findings of this study are available on request from the corresponding author (R. Xie) upon reasonable request.

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
