# Peer review of "Temporal and Spatial Dynamics of Organ Water Content in Maize with Different Senescence Types"

_plants, 2023, doi:10.3390/plants12183269_

Round 1

Reviewer 1 Report

The manuscript, entitled "  Temporal and Spatial Dynamics of Organs Water Content after Silking in Maize with Different Senescence Types " This work is merited for publication in Plants after some major modification. So, I have some points that may help to improve the work as follows:

 1-Abstract is good but need more explain about the main aim of work

 2- The introduction should be extended to discuss the hypothesis and research questions in details. Additionally, the introduction should cover the recent literature related to this subject.

 3- Material and methods

The methodologies should be explained in details so that the results are reproducible.

4-Results

The results are clear and important.

 5-Discussion

The discussion section still needs improvement, and should be linked to the findings of the previous reports on this topic.

6- The conclusion

A section for conclusions need more explain and should include the most significant findings and future works only.

7- English writing should be checked by a native English-speaking expert.

English writing should be checked by a native English-speaking expert.

Minor editing of English language required

Author Response

NOTE: Generally, the revision parts to Reviewers were marked with Red words in the manuscript. In addition, we have invited a professional editing service to revise the manuscript, and the revision parts were  marked with black fonts.

The comments of reviewer and our responses were as follow:

Q1-Abstract is good but need more explain about the main aim of work

R: Thank you for your suggestion. We have added some statements about the main aim of work in Abstract to make it more logical.

Q2- The introduction should be extended to discuss the hypothesis and research questions in details. Additionally, the introduction should cover the recent literature related to this subject.

R: Thank you for your suggestion. In new manuscript, we have further extended the hypothesis and scientific questions in the introduction.

The physiological and functional were different between the reproductive organs (kernels) and vegetative organs (stalks and leaves) of plants. We think that, as a plant, it is essential to maintain the stability of reproductive growth and should not be restricted by some characteristics of vegetative organs. The “stay-green” of leaf and stalk or delayed senescence can be inherited as genetic diversity, but should not affect the formation of maize offspring, as this can be fatal to the species. Therefore, we proposed a hypothesis that the water dynamics of vegetative organs such as stalks and leaves would not affect the kernels of vegetative organs. We also confirmed this hypothesis in this study. In addition, we have also added some physiological research of water in the introduction.

We hope these revised could address your concerns.

Q3- Material and methods:

The methodologies should be explained in details so that the results are reproducible.

R: Thank you for your suggestion. We have revised some statement about the sampling and measurement method in Material and methods.

Q4-Results:

The results are clear and important.

R: Thank you for your approval.

Q5-Discussion:

The discussion section still needs improvement, and should be linked to the findings of the previous reports on this topic.

R: Thank you for your suggestion. We have added some descriptions of physiological aspects in the discussion, and tried to explain our findings based on previous research. We have also deleted and rewritten some parts of the discussion.

Q6- The conclusion:

A section for conclusions need more explain and should include the most significant findings and future works only.

R: Thank you for your suggestion. We have revised the conclusion in manuscript.

Q7- English writing should be checked by a native English-speaking expert.

R: Thank you for your suggestion. We have invited a professional editing service to revise the manuscript.

Reviewer 2 Report

MAJOR ISSUES

1.    Authors should include the missing information i.e., research gaps and the significance of the research.

2.    Why these cultivars were selected (9 in number)? Was there any significance in selecting the same? If so the authors needs to justify.

3.    Why the authors selected two different locations, if the cultivars were inter-changed with the selected places were the results obtained are similar?

4.    I want to advise adding new pieces of information about biochemical and molecular mechanisms. To improve the MS, authors could also involve new aspects of regulation mechanisms

5.    In line 539-540: if the water content didn’t affect the kernals, than what is the use of this study? Hence authors should give an explanation on the significance of the work carried out.

MINOR ISSUES

1.    Introduction should be cut short and should include novelty of the proposed work.

2.    Do a Scopus database and include in the review the newest information available. Avoid as many possible references older than 10 years.

3.    Authors should look into the existing overlapped descriptions throughout the MS.

4.    English level is generally acceptable. Nevertheless, I recommend that the authors read all sections carefully to improve it as I found some unclear passages and grammar mistakes before publication. Provide the full form of abbreviations wherever cited first.

5.    Format all the references in the same manner; there are some references in which the whole text part is not equally spaced

6.    Results and Discussion part need improvement.

7.    Conclusion should highlight the significant findings of the study.

English level is generally acceptable. Nevertheless, I recommend that the authors read all sections carefully to improve it as I found some unclear passages and grammar mistakes before publication. Provide the full form of abbreviations wherever cited first.

Author Response

NOTE: Generally, the revision parts to Reviewers were marked with Red words in the manuscript. In addition, we have invited a professional editing service to revise the manuscript, and the revision parts were  marked with black fonts.

The comments of reviewer 2 and our responses were as follow:

MAJOR ISSUES

  1. Authors should include the missing information i.e., research gaps and the significance of the research.

R: Thank you for your suggestion. According to your suggestion, we proposed our research hypothesis in introduction and added some statement. For the relationship of maize organs water dynamic, we have further discussed and extended in the revised manuscript. We think that, as a plant, it is essential to maintain the stability of reproductive growth and should not be restricted by some characteristics of vegetative organs. The “stay-green” of leaf and stalk or delayed senescence can be inherited as genetic diversity, but should not affect the formation of maize offspring, as this can be fatal to the species. Therefore, we proposed a hypothesis that the water dynamics of vegetative organs such as stalks and leaves would not affect the kernels of vegetative organs.

We hope that these revised could help readers better understand the significance of this work and address your concern.

  1. Why these cultivars were selected (9 in number)? Was there any significance in selecting the same? If so the authors needs to justify.

R: We strongly agree with your suggestion. The selection of cultivars is very important for the universality of the study. In this study, we selected 12 cultivars with different growth period, of which 9 were from summer maize and 3 were from spring maize. The cultivars at both two sites included early, medium and late growth period. We were concerned that cultivars with different period may contributed to different results. So, we tried to test as many different scenarios as possible to reduce the error of the test. The growth period of the cultivars used in this study basically included all the range of growth period in production, and their growth period can also be found in the manuscript. At the same time, some cultivars were also commonly used by many researchers, such as Zhengdan958, Xianyu335 and Dika517. We hope these cultivars to get more reliable and acceptable result for many researchers.

  1. Why the authors selected two different locations, if the cultivars were inter-changed with the selected places were the results obtained are similar?

R: As mentioned earlier, we tested two sites, but they had very great difference at growing environments. One was from the spring planting area and the other was from the summer planting area. Because spring and summer maize had different growth period, the final yield and plant shape. In order to obtain more general results, we cannot ignore the difference of spring maize and summer maize.

  1. I want to advise adding new pieces of information about biochemical and molecular mechanisms. To improve the MS, authors could also involve new aspects of regulation mechanisms

R: Thank you for your suggestion. Leaf senescence and kernel development involve many biochemical, molecular and gene-related mechanisms and regulatory mechanisms, especially the relationship between leaf water and kernel water in this study. Although we have not been able to observe the link between the water dynamic of leaf and kernel at the apparent level, there are certainly many interesting physiological or molecular mechanisms between them. That could be a worthwhile subject. We will pay more attention to this aspect in future research. Thank you again for your advice!

  1. In line 539-540: if the water content didn’t affect the kernels, than what is the use of this study? Hence authors should give an explanation on the significance of the work carried out.

R: We agreed your concern. In conventional logic, if the leaves and stalks showed a delayed senescence or a delayed water loss, it would be natural to assume that the kernels would show a similar phenomenon. However, apparently, the water loss process of kernels was not affected by stalks and leaves in our results. Unfortunately, we can't give a detailed reason at the physiological or genetic level now. We think that in terms of physiological function, the kernel is a part of the reproductive organ and has a strict growth rhythm. The development and dehydration of the maize kernels should follow the established regulation under normal growth conditions. The leaves showed the state of delayed water loss, which may be related to the functional redundancy of the organs. We guessed that there may be no direct regulatory mechanism between them. At least, leaf water content did not show absolute regulation of the kernel development process in the today’s cultivars tested in this work.

However, from the actual agricultural production, it is convenient to find a simple indicator to predict the kernel water content in order to determine the harvest time of maize. Leaf (or canopy) water content is a relatively easy indicator to obtain because it can already be detected using remote sensing. In order to better improve the prediction accuracy, it is necessary to clarify the normal, actual, and detailed water dynamics of kernels or leaves, but there are few researches in this part. In addition, in the long time scale, the water content of both leaves and kernels showed the same and decreasing trend. However, according to the results of this study, the direct prediction of kernel water content using leaf water content will inevitably have a large error due to the differences in the senescence and stay-green of maize cultivars. Therefore, the proposed results may force more researchers to pay attention to this problem, which has certain significance for the development of quantitative remote sensing and the improvement of remote sensing accuracy in the future.

MINOR ISSUES

  1. Introduction should be cut short and should include novelty of the proposed work.

R: Thank you for your suggestion. We have simplified and revised the Introduction.

  1. Do a Scopus database and include in the review the newest information available. Avoid as many possible references older than 10 years.

R: Thank you for your suggestion. We have removed some old references in the manuscript.

  1. Authors should look into the existing overlapped descriptions throughout the MS.

R: Thank you for your suggestion. We have deleted and rewritten some parts in the new manuscript.

  1. English level is generally acceptable. Nevertheless, I recommend that the authors read all sections carefully to improve it as I found some unclear passages and grammar mistakes before publication. Provide the full form of abbreviations wherever cited first.

R: Thank you for your suggestion. We have invited a professional editing service to revise the manuscript. The full form of abbreviations have revised in the new manuscript.

  1. Format all the references in the same manner; there are some references in which the whole text part is not equally spaced

R: Thank you for your suggestion. We have revised it.

  1. Results and Discussion part need improvement.

R: Thank you for your suggestion. We have revised some descriptions in the results section. In addition, we have added some descriptions of physiological aspects in the discussion, and tried to explain our findings based on previous research. We have also deleted and rewritten some parts.

  1. Conclusion should highlight the significant findings of the study.

R: Thank you for your suggestion. We have revised the conclusion in manuscript.

Reviewer 3 Report

In present study, the temporal and spatial dynamics of water content in stalks and leaves were studied based on maize cultivars with different growth period. The decline of leaves water content started from the bottom, then from the bottom and top to the middle, and the leaves water content around ear or 1-2 leaves above ear decreased at the last, while the water content of stalk internodes decreased one by one from top to bottom. Spatially, the water content of maize leaves and stalks showed a decreasing vertical water gradient from bottom to top. At the filling stage, the vertical water gradients of stalk and leaf decreased 0.99% and 1.27% per position from bottom to top. In time course, the water dynamics of leaves and internodes could be divided into three main stages: the slow loss stage, the rapid loss stage and the balance stage. All three stages were fitted by linear equation. There was an important water threshold from slow loss stage to rapid loss stage, which represented the irreversible turning point of organ death. Water content at this turning point was stable across maize cultivars, with the minimum threshold of about 60%. In addition, when the normalized growth period was used, the water dynamics of maize organs can be divided into two types: stay-water type and general type. So the method of normalized growth period may also have potential in evaluating the stay-green of maize cultivars. The stay-water cultivars had a longer water duration and delayed senescence time than the general cultivars, and if maize leaves were stay-water, the stalks were also stay-water correspondingly. In addition, the water loss rate of general type was about twice than stay-water type in rapid loss stage. Therefore, in the process of breeding maize cultivars, it may be not robust to evaluate the dehydration of kernels by the stay-green of maize leaves. The manuscript is interesting, and I have just minor remarks that should be addressed.

Determination of organs water content of maize plants - the dry weights were measured after drying at 105 ºC for 30 minutes and then drying at 85 ºC for at least 48 hours in air oven… Why the authors selected these conditions for drying? Did they consider that drying ant higher temperature could cause thermal degradation present natural organic compounds and some formed volatiles or present volatiles can evaporate that could influence the measurement of water loss!

At the end of introduction part there is need to clearly state what is the novelty of present research. What was done for the first time? Although the objectives of our study were stated as: (1) to clarify the vertical distribution of water content in leaves and stalks of maize; (2) to decipher the water dynamic process of each leaf and stalk internode with the process of senescence; (3) to evaluate the water relationship between stalk/leaf and ear of maize, it is not clear what is the novelty and contribution of present paper to the overall knowledge and what was done for the first time. Numerous questions were raised in lines 89-95, but it is necessary to point out the novelty better. Also the research hypothesis should be written and how it is to be tested.

-

Author Response

NOTE: Generally, the revision parts to Reviewers were marked with Red words in the manuscript. In addition, we have invited a professional editing service to revise the manuscript, and the revision parts were  marked with black fonts.

The comments of reviewer 3 and our responses were as follow:

  1. Determination of organs water content of maize plants - the dry weights were measured after drying at 105 ºC for 30 minutes and then drying at 85 ºC for at least 48 hours in air oven… Why the authors selected these conditions for drying? Did they consider that drying ant higher temperature could cause thermal degradation present natural organic compounds and some formed volatiles or present volatiles can evaporate that could influence the measurement of water loss!

R: Thank you for your suggestion. Considering that fresh living plant samples might lose organic matter due to respiration, we first dried them at a high temperature (105 °C) to inactivate the enzymes, and then the samples were placed in an oven at 85 °C for 48h. This temperature and time could remove most of water in sample according to our previous study (ref. Maize grain moisture content correction: From nonstandard to standard system. Biosystems Engineering). In addition, we agreed with your concern that high temperatures may indeed cause the degradation of some compounds, which was inevitable. However, due to structural differences, the drying time required for stalks, leaves and kernels etc. will be very different. If a lower temperature was used, this may further increased the measurement error. In order to allow all organs to remove as much free water from the tissues as possible under a uniform standard, we used 85 °C to drying. This temperature was also the popular temperature used in the measurement of dry matter and kernel water content and it may be likely to be accepted by most researchers.

  1. At the end of introduction part there is need to clearly state what is the novelty of present research. What was done for the first time? Although the objectives of our study were stated as: (1) to clarify the vertical distribution of water content in leaves and stalks of maize; (2) to decipher the water dynamic process of each leaf and stalk internode with the process of senescence; (3) to evaluate the water relationship between stalk/leaf and ear of maize, it is not clear what is the novelty and contribution of present paper to the overall knowledge and what was done for the first time. Numerous questions were raised in lines 89-95, but it is necessary to point out the novelty better. Also the research hypothesis should be written and how it is to be tested.

R: Thank you for your suggestion. According to your suggestion, we have reframed the scientific questions and hypothesis in the Introduction and added some statement of hypothesis and novelty of research. We also further elaborate the extended implications of these scientific questions. We hope that these revised could help readers better understand the significance of this work.

Round 2

Reviewer 1 Report

Authors have suitably revised the manuscript by addressing the reviewer comments and suggestions. This can be accepted for publication.

Minor editing of English language required

Reviewer 2 Report

Accept

Accept